# Macrophage-Based Therapeutic Strategies in Hematologic Malignancies

**DOI:** 10.3390/cancers15143722

**Published:** 2023-07-22

**Authors:** Saeed Khalili, Fatemeh Zeinali, Atousa Moghadam Fard, Seyed Reza Taha, Andarz Fazlollahpour Naghibi, Kimia Bagheri, Mahdieh Shariat Zadeh, Yeghaneh Eslami, Khashayar Fattah, Naghmeh Asadimanesh, Armin Azarimatin, Bahman Khalesi, Faezeh Almasi, Zahra Payandeh

**Affiliations:** 1Department of Biology Sciences, Shahid Rajaee Teacher Training University, Tehran 1678815811, Iran; s.khalili88@gmail.com; 2Department of Immunology, Faculty of Medicine, Ahvaz Jundishapur University of Medical Sciences, Ahvaz 6135715794, Iran; zeinali_fatemeh1998@yahoo.com; 3Universal Scientific Education and Research Network (USERN), Tehran 4188783417, Iran; atoosa.mf74@gmail.com; 4Oncopathology Research Center, Iran University of Medical Sciences, Tehran 1449614535, Iran; reza.taha.rt7@gmail.com (S.R.T.); mahdiehshariatzadeh@gmail.com (M.S.Z.); 5Infectious Diseases and Tropical Medicine Research Center, Health Research Institute, Babol University of Medical Sciences, Babol 4717641367, Iran; andarz.fpr@gmail.com (A.F.N.); bagheriikimia@gmail.com (K.B.); 6Faculty of Medicine, Mazandaran University of Medical Sciences, Sari 4815733971, Iran; yeganeslami99@gmail.com; 7School of Medicine, Shahid Beheshti University of Medical Sciences, Tehran 1985717411, Iran; khashayarfatah@gmail.com (K.F.);; 8Department of Veterinary Medicine, Shabestar Branch, Islamic Azad University, Shabestar 5381637181, Iran; azarimatinarmin@yahoo.com; 9Department of Research and Production of Poultry Viral Vaccine, Razi Vaccine and Serum Research Institute, Agricultural Research, Education and Extension Organization, Karaj 3197619751, Iran; khalesi20022002@yahoo.com; 10Pharmaceutical Biotechnology Lab, Department of Microbial Biotechnology, School of Biology and Center of Excellence in Phylogeny of Living Organisms, College of Science, University of Tehran, Tehran 1416634793, Iran; 11Department of Molecular Biosciences, Wenner-Gren Institute, Stockholm University, SE 106 91 Stockholm, Sweden

**Keywords:** macrophages, hematologic malignancies, tumor microenvironment

## Abstract

**Simple Summary:**

Tumor-associated macrophages (TAMs) are the most prevalent immunosuppressive myeloid cells in the tumor microenvironment, playing significant functions in the regulation of tumor progression, invasion, and metastatic processes. The M1 and M2-polarized phenotypes of TAMs (immunostimulatory and immunosuppressive myeloid cells, respectively) have been potentially implicated in various cancers and autoimmune diseases. Understanding the precise function of TAMs could improve the assessment of the cancer response to T cell-based treatments and reverse tumor resistance to conventional therapies. Here, the involvement of TAMs in the development of various cancers, mainly hematologic tumors, and their pleiotropic activities are comprehensively discussed.

**Abstract:**

Macrophages are types of immune cells, with ambivalent functions in tumor growth, which depend on the specific environment in which they reside. Tumor-associated macrophages (TAMs) are a diverse population of immunosuppressive myeloid cells that play significant roles in several malignancies. TAM infiltration in malignancies has been linked to a poor prognosis and limited response to treatments, including those using checkpoint inhibitors. Understanding the precise mechanisms through which macrophages contribute to tumor growth is an active area of research as targeting these cells may offer potential therapeutic approaches for cancer treatment. Numerous investigations have focused on anti-TAM-based methods that try to eliminate, rewire, or target the functional mediators released by these cells. Considering the importance of these strategies in the reversion of tumor resistance to conventional therapies and immune modulatory vaccination could be an appealing approach for the immunosuppressive targeting of myeloid cells in the tumor microenvironment (TME). The combination of reprogramming and TAM depletion is a special feature of this approach compared to other clinical strategies. Thus, the present review aims to comprehensively overview the pleiotropic activities of TAMs and their involvement in various stages of cancer development as a potent drug target, with a focus on hematologic tumors.

## 1. Introduction

Macrophages are among the major cellular components which are involved in numerous tumors. They have remarkable functions in the promotion of tumorigenesis in the tumor microenvironment (TME) through the facilitation of angiogenesis, invasion, metastasis, and immunosuppression [1]. Tumor-associated macrophages (TAMs) are the most prevalent immune-related cells in the TME. They play a substantial role in tumor progression and metastatic processes through various mechanisms [2,3]. Poor survival and a high rate of infiltration are the remarkable properties of macrophages that indicate them as promising targets of anticancer therapies. The efficacy of TAM targeting has already been confirmed in numerous clinical trials [4,5]. Furthermore, the combination of macrophage-directed therapies with other therapies (such as chemotherapies and immunotherapies) has shown complementary effects. Thus, devising novel combined therapeutic strategies requires a good grasp of TAM biology and its intricate interplay with the TME [6].

In this review, we aimed to discuss the diverse roles of macrophages in different cancer development pathways, including cancer initiation, promotion, invasion, metastasis, and angiogenesis. The role of TAMs as diagnostic and prognostic biomarkers is also outlined. The promising application of TAM-based approaches in the treatment of malignancies is also discussed. Moreover, the role of macrophages in hematologic cancers, like acute lymphoblastic leukemia (ALL), acute myeloid leukemia (AML), and chronic lymphocytic leukemia (CLL), is comprehensively discussed. These cancers are characterized by the presence of leukemia-associated macrophages (LAMs).

## 2. Diversity, Polarization, and Function of TAMs

Macrophages are key immune cells that are derived from monocytic progenitors in the bone marrow and are known as tumor-associated macrophages (TAMs) [5]. Aside from their role in the initiation of inflammatory responses against a stimulus, homeostasis, and the elimination of unnecessary cells, macrophages play various roles in cancer development. The highly plastic macrophages can undergo remarkable changes in their function in response to cues in the TME [7]. In established malignancies, poor prognosis or tumor progression is often strongly associated with a high macrophage infiltration in different tumors, including glioma [8], melanoma [9], breast [10], bladder [11], and prostate cancer [12]. Conversely, high macrophage infiltration is related to better prognosis in colorectal and gastric cancers [13]. This certain discrepancy could be rooted in the functional and phenotypical heterogeneity of macrophages in different types of tumors. TAMs can be broadly classified into two subsets based on their functions. M1-like TAMs (pro-inflammatory and anti-tumor) and M2-like TAMs (anti-inflammatory and pro-tumor) are the major subsets [14]. Lipopolysaccharides, interleukin-1, tumor necrosis factor, and/or granulocyte-macrophage colony-stimulating factor (M-CSF) can actuate the M1-like TAMs. These TAMs can detect and eliminate cancer cells through phagocytosis and cytotoxicity, and initiation of anti-tumor immunity via pro-inflammatory cytokines [15]. On the other hand, M2-like TAMs, which are alternatively activated and induced by various factors within the local microenvironment, can promote tumor growth and TME remodeling through the production of immunosuppressive factors, growth factors, proteases, and pro-angiogenic molecules [16]. The expression of inducible nitric oxide synthase (iNOS) and arginase 1 (ARG1) is suggested to be involved in the intrinsic regulation of macrophage polarization, which could lead to the activation of M1 and M2 macrophages. Tricarboxylic acid (TCA), glutamine (due to its ability to refill TCA cycle metabolites), and serine (that feeds into the one-carbon metabolism) are also involved in macrophage polarization. The amino acids corresponding metabolic pathways, and probable mechanisms of TAMs polarization, are comprehensively discussed by Kieler et al. [17]. Overall, phenotypic switches in TAMs depend on combinational microenvironmental factors, including the action of hypoxia and the availability of cytokines. In addition, the lack of an approved strategy for intratumoral hypoxia measurement remains the main obstacle to a better characterization of hypoxic TAMs. For example, the M1 phenotype is generally generated in LPS-mediated hypoxic responses, while hypoxic TAMs are more strongly associated with an M2-like response [18]. These properties emphasize the necessity of considering hypoxic stress in the context of tumor immunotherapy.

## 3. Pleiotropic Activities of TAMs in Tumors

Macrophages are specialized to function in specific microenvironments, which contributes to their location in tumor tissues [19]. The tumor stroma, the tumor center, and the boundary between the tumor cells and the stroma, which is called the invasive front, are the three main locations where macrophages can be found [20,21]. It has been indicated that the distribution pattern and distinct functions of macrophages may be related to different cancer progression mechanisms and the location-related signals they receive [22]. For example, in colorectal cancers, M2 phenotype macrophages may be preferentially involved in promoting the movement of cancerous cells in the invasion zone, facilitating metastasis in stromal and perivascular areas, and stimulating angiogenesis in avascular and peri-necrotic hypoxic areas via the induction of S100A8 and S100A9 (calcium-binding proteins correlated with differentiation and metastasis) [23]. Otherwise stated, the distribution pattern of macrophages may be linked to different mechanisms of cancer development. Notably, some gastric cancer cases have been characterized by a stroma-dominant pattern, leading to greater malignancy. This property could be engrained into the aggregation of macrophages in the tumor stroma. It may also contribute to the remodeling of the extracellular matrix (ECM) and stroma activation in conjunction with other elements of the stromal compartment such as matrix metalloproteinase 9, lysyl oxidase, and type IV collagen [24]. Moreover, the ratio of CD163^+^ to CD68^+^ macrophages on the invasive front of colorectal cancer has been suggested as a potential prognostic marker. Future studies could focus on exploring the relationship between the varied morphologies of macrophages, tumor positions, and their role in different distribution patterns [25].

## 4. Macrophages and Cancer Development

The frequent presence of TAMs is often associated with insignificant clinical outcomes in most tumors [5,26], influencing the relapse of tumors after conventional cancer treatments. TAM targeting has garnered a lot of interest as a potential therapeutic strategy and several therapeutic agents that specifically target these cells have been tested in clinical trials. In a recently published review that investigated 300 studies, an obvious relation was identified between the infiltration of macrophages (M1 or M2 subtypes) and the prognosis of various solid cancer types. Specifically, the attending of M2-subtype macrophages was related to a poor outcome, while the presence of M1-subtype macrophages was associated with a favorable prognosis [27]. Hitherto, immune suppression, angiogenesis, chronic inflammation, and invasion/metastasis are the well-characterized tumor-promoting mechanisms of TAMs [5].

### 4.1. Tumor-Progressing Inflammation and Macrophages

In a healthy state, inflammation is a response to external factors that helps to restore homeostasis [28]. However, chronic inflammation can increase the risk of carcinogenesis. Before the formation of tumors, tumor-promoting inflammation can occur and support tumor growth through suppression of the immune system, promotion of neoangiogenesis (the generation of new vascular networks to supply cancer cells), and oncogenic mutations [29]. Cell death within the tumors could lead to the liberation of damage-associated molecular patterns (DAMPs), like high mobility group box 1 (HMGB1), heat shock proteins (HSPs), or ATP [5,30]. The released DAMPS can activate macrophages and dendritic cells, causing anti-tumor immunity stimulation. Chronic stimulation can result in immunosuppression via the elevated production of interleukin-10 (IL-10), which suppresses the expression of pro-inflammatory cytokines and promotes the formation of regulatory T cells (Tregs) (Figure 1) [31]. In addition, macrophages can contribute to tumor-promoting inflammation through the secretion of immunostimulatory cytokines, such as interleukin-6 (IL-6) [32], and tumor necrosis factor-alpha (TNFα) [33]. These cytokines can stimulate the immune response, backing the tumor growth and survival of cancerous cells [34]. TNFα activates the nuclear factor kappa-B (NF-κB) pathway upon binding to TNFR1/2 receptors. TNFRs (tumor necrosis factor receptors) are membrane proteins that activate cell death. Activation of this pathway could lead to the control of target gene expression (e.g., vascular endothelial growth factor (VEGF) and IL-6) and the stimulation of neo-angiogenesis. IL-6 could then promote cell proliferation and differentiation [35] via the JAK/STAT3 pathway [36].

Another tumoricidal function of TAMs is the blockade of macrophage recruitment by the macrophage migration inhibitor factor (MIF), which is induced via phagocytosis [37] and the propagation of TNFα and interleukin-1 beta (IL1β) [38]. Moreover, the TAM-secreted interleukin-18 and 22 can increase the production of IFNγ and IL-2, which could lead to the increased cytotoxic activity of natural killer cells [39]. The significance of TAMs in tumor development has been highlighted in numerous studies [40]. For instance, the genetic ablation of the Csf1 gene (which encodes M-CSF and is required in macrophage maturation) could lead to the delayed metastasis of mammary carcinoma, whereas the transgenic expression of the M-CSF could speed up pulmonary metastasis [41]. Analogous results have been recognized in thyroid and osteosarcoma cancer cells [42]. These outcomes indicate the existence of an intricate balance between the tumor-promoting or killing functions of TAMs. 

### 4.2. Angiogenesis

Macrophages are essential in the development of cancer due to their ability to promote angiogenesis. Their presence is often correlated with increased blood vessel density in the tumor microenvironment [43]. To support the swift proliferation of malignant cells, the tumor requires a high supply of nutrients and oxygen which are delivered through a capillary network formed during angiogenesis [44]. The released growth factors in the tumor microenvironment are responsible for the regulation of this process. However, the obligation and structure of the newly made vascular tissues are often abnormal due to poor regulation. This property leads to the elevated permeability of vessels and connections to disease development. Considering the elevated rate of cell death in tumor tissues, TAMs are attracted to hypoxic areas to stimulate the formation of new blood vessels [45]. The transcription factor HIF1α (hypoxia-inducible factor α) is an oxygen-dependent transcriptional activator consistently observed in macrophages. It plays critical roles in the angiogenesis of tumors, regulates the response to hypoxic stress (by switching from aerobic to anaerobic metabolism), and induces CCL2, CXCR4, and endothelin expression as HIF1(hypoxia inducible factor-1) target genes [46], which could lead to macrophage recruitment into tumors [47]. In addition, the process of neo-angiogenesis is adjusted by various elements which are produced by TAMs, including platelet-derived growth factor (PDGF), vascular endothelial growth factor (VEGF), matrix metalloproteinases (MMPs), and angiopoietin-1 [48]. These factors play a role in the proliferation and maturation of endothelial cells, the chemotaxis of macrophages and ECs [48], and the breakdown of the extracellular matrix (to allow for the formation of novel vascular sprouts) [49]. PDGF released by TAMs and platelets could promote the infiltration of pericytes, which are important for vessel maturation and remodeling [44]. They can also release Angiopoietin-1, which helps to stabilize newly formed vessels by binding to the Tie-2 receptor on endothelial cells. Tie 2 is a tyrosine kinase receptor with a critical function in vascular stability. Tie-2 receptor-expressing monocytes (TEMs) are responsible for enhancing the blood vessel formation of macrophages in tumors and may act as precursors of proangiogenic TAMs [50]. Therapeutic targeting of these pathways may be a potential approach for cancer treatment (Figure 2) [5].

### 4.3. Role of Macrophages in Tumor Cell Invasion and Metastasis

Tumor cell invasion and metastasis are responsible for failure in cancer treatment and the great number of cancer-related deaths [51,52]. Based on previous pieces of evidence, M2 TAMs can increase the growth, invasion, and metastasis of tumor cells and stimulate angiogenesis; on the other hand, M1 TAMs can provoke anti-tumor effects by producing and secreting pro-inflammatory cytokines and exerting macrophage-mediated cytotoxicity [2,3,53,54,55,56]. Overall, TAM inhibition is considered a promising cancer treatment strategy [54,57]. Yang et al. [54] found an increased infiltration of TAMs, especially M2 macrophages, in both the peritumoral and intertumoral sites of the solid pseudopapillary pancreatic tumor with metastatic features compared to the patients with capsular features [54]. In contrast, Konstantinov et al. [58] found no direct associations between M2 macrophages with metastatic behavior in colorectal cancer. Their results suggest that M2 macrophages could restrict the metastatic processes [58]. TAMs play a substantial role in the metastatic process by contributing to invasion, angiogenesis, intravasation, extravasation, colonization, survival of tumor cells, induction of hypoxia, and pre-metastatic niche formation [2,56,59]. TAMs could suppress CD8⁺ T cell responses by the secretion of interleukin-10 (IL-10) and promote the differentiation of naive CD4⁺ T cells into Treg [57,60]. Activated TAMs produce and secrete various soluble factors, such as tumor-transforming growth factor-β (TGF-β), necrosis factor-α (TNF-α), IL-1β, and IL-8. These factors could ultimately damage the basement membrane of tumor endothelial cells and facilitate epithelial-mesenchymal transition (EMT) processes, which could promote invasion [3,55,61,62]. Furthermore, TAMs, especially M2 macrophages, are capable of extracellular matrix degradation and help tumor cell migration by secreting proteolytic enzymes, including matrix metalloproteinases (MMPs, such as MMP9, MMP7, and MMP2), cathepsins, and serine proteases [3,61,63]. Furthermore, TAMs can enhance invasion, migration, and the circulating tumor cell-mediated metastasis of colorectal cancer by regulating the JAK2/STAT3/miR-506-3p/FoxQ1 axis, which leads to CCL2 production [64]. M2 macrophages also secrete chitinase-3-like protein 1 (CHI3L1), which could promote gastric and breast cancer metastasis by the initiation of the mitogen-activated protein kinase (MAPK) signaling pathway [65]. In addition, TAMs promote tumor angiogenesis through secreting pro-angiogenic factors such as fibroblast growth factor (FGF) and vascular endothelial growth factor (VEGF), which facilitate metastasis [2,60]. Intravasation and extravasation of tumor cells are critical steps in metastasis, which are both promoted by TAMs [3]. Triggering the PI3K/Akt survival pathway through engaging vascular cell adhesion molecule-1 (VCAM1) and the secretion of cytokines and chemokines can increase the survival of cancer cells [3,66,67,68,69,70]. In addition, cat eye syndrome chromosome region candidate 2 (CECR2) is an epigenetic regulator which is necessary for breast cancer metastasis [71]. Zhang et al. [71] found an association between CECR2 expression and increased M2 TAMs in the TME, which promotes breast tumor metastasis. They found that CECR2 promotes breast cancer metastasis by regulating M2 TAMs [71].

The pre-metastatic niche (a well-prepared environment in a secondary organ for the colonization of tumor cells) is a specialized environment, including infiltrating immune cells, tumor cells, activated stromal cells, and extracellular matrix regulating tumor progression, while the pre-metastatic niche is an essential requirement for the colonization of circulating pro-tumor cells in a particular organ and metastasis of primary tumors. Macrophages are one of the most remarkable immune cells within pre-metastatic niches [72]. Zhao et al. [73] found that polarized M2 TAMs could induce the formation of pre-metastatic niches. Macrophage depletion could significantly decrease the number of metastatic nodules, which portrays the substantial role of macrophages in tumor metastasis [72,74,75]. Recent studies have evaluated TAM-based cancer treatment; chimeric antigen receptor-macrophage (CAR-M) therapy is a promising cancer treatment with significant outcomes [76,77]. CAR-M is a cancer therapy option in which M2 macrophages are manipulated into the M1 phenotype; as previously pointed out, M1 macrophages have anti-tumor and pro-inflammatory effects [76,78]. According to recent studies, CAR-M therapy could increase overall survival and suppress tumor growth by converting M2 into M1 macrophages, which express pro-inflammatory cytokines and chemokines, resist the effect of immunosuppressive chemokines, present antigens to T cells, upregulate antigen presentation machinery, and prevent metastasis [76,79,80].

The stimulation of cancer cell motility by Wnt5a [81], expression of MMPs via SPARC/Osteonectin, and adjustment of collagen fibers are the other agents causing macrophage-mediated tumor invasion [82,83]. It has been mentioned that activation of the CCL2/CCR2 axis could facilitate the leakage of cancer cells to other areas [83], enhance the secretion of MAM (metastasis-associated macrophage)-derived CCL3, and promote the bone metastasis of prostate cancer [84]. Bone destruction through the activation of osteoclasts could trigger the release of tumor growth factors [85], while the blockade of CCL2 with specific shRNA or neutralizing antibodies could significantly impair bone resorption and prostate cancer-induced formation of osteoclasts [86]. Moreover, the interaction between α4-integrin (expressed by MAMs) and vascular cell adhesion protein 1 in cancer cells could increase lung tumor development [67]. Given these circumstances, the multidimensional function of macrophages could be determined in the metastatic process via clear and explicit evidence.

### 4.4. TAMs as Diagnostic and Prognostic Target

TAMs have been demonstrated to act as potential biomarkers of cancer stage and progression. Research has demonstrated that the concentration of TAMs in the tumor stroma can predict the size, stage, and metastasis of various tumors. This property could lead to a more accurate prognosis and personalized treatment planning for cancer patients. Patients with higher levels of TAMs have a lower overall survival rate compared to those with lower levels of TAMs. As such, TAMs may be useful for risk assessment, early diagnosis, and prognosis in cancer patients [87]. The quantification of infiltrated TAMs as an essential diagnostic target in various tumors can be accomplished through different morphological methods, cell-surface marker profiling, and gene expression analysis [88]. Although TAMs are mainly recognized as CD68 positive, the AAM (alternatively activated macrophages or alternatively activated M2 macrophages) endotype is distinguished with CD163, CD206, and CD204, while the CAM endotype is distinguished with CD40 [89] and HLA-DR expression [90]. An indication of advanced cancer stages [88], high macrophage density is known as a prognostic marker to estimate chemotherapy results and survival [91]. For instance, failure in Hodgkin lymphoma treatments is correlated with overexpressed macrophages in lymph nodes [92]. Moreover, the prevention of tumor progression through TAM re-education has been considered a clear determinant of the efficiency of postsurgical chemotherapy in pancreatic cancer [88]. Therefore, TAM quantification could be considered a useful approach in patients who are more responsive to chemotherapy.

## 5. TAMs as a Therapeutic Target

Considering the dual function of TAMs in tumor microenvironments, they have appeared as promising therapeutic targets. Various therapeutic strategies have focused on TAM targeting, aiming to reprogram, deplete, or adjust any TAM-secreted mediators. Early clinical trials have suggested that targeting the checkpoints of myeloid cell function as negative regulators could bear antitumor potential. Macrophages are proper candidates for cell therapy due to the continuous recruitment of myelomonocytic cells into tumor tissues [93]. Overall, ongoing available implements in the oncology armamentarium could be complemented and synergized with macrophage-centered therapeutic strategies (Table 1). The widely applied TAM-based strategies include conventional anticancer therapies (recruitment, repolarization, and depletion), immune checkpoint blockade (ICB), vaccination, cell therapy, and the administration of apoptotic peptides and nanoparticles targeting TAMs.

### 5.1. TAMs in Conventional Cancer Therapies

Some chemotherapy drugs can stimulate the dissemination of cancer-related molecules through a process called immunogenic cell death. Macrophages are engaged through this process in a beneficial immune response against cancer [176]. Other cancer treatment strategies target macrophages by a reduction in their numbers, like the treatment of ovarian cancer and certain types of sarcomas with an approved marine-derived compound named Trabectedin. The depletion of TAMs is required for the full anti-tumor effect of trabectedin [102]. Some anti-cancer drugs can also change the polarization of TAMs. This change could lead to an increased responsiveness to treatment, such as 5-fluorouracil in colorectal cancer, gemcitabine in pancreatic cancer [88], and high-grade ovarian cancer treated with platinum-based neoadjuvant [177]. Platinum-based neoadjuvant can lead to DNA damage through the production of reactive oxygen species (ROS). In addition, the gut microbiota can stimulate the production of ROS with intratumoral phagocytes, therefore, the effectiveness of these compounds can be enhanced [178].

Since total macrophage depletion is not clinically bearable for a long time [16], strategies have been developed to suppress macrophages via antibodies [126] or small molecule compounds [179]. These strategies involve macrophage targeting with antibodies (such as anti-CSF1R) or small compounds (such as bisphosphonates). These targeting antibodies and molecules inhibit the recruitment of macrophages, deplete their number, and re-educate them [180]. Tumor-infiltrating leukocytes, including TAMs, are fundamental players in the antitumor activity of certain monoclonal antibodies (mAbs). These mAbs trigger the FcγR expressing immune cells to kill tumor cells and perform phagocytosis. The currently prescribed antibodies include rituximab [181], cetuximab [182,183], trastuzumab [184], and daratumumab. Frequently, there is a relation between the density of TAMs and vessels in tumor tissues, which is due to the active responsiveness of TAMs to angiogenic growth factors. VEGF is primarily among these growth factors [185]. Therefore, TAMs modulate the efficiency of antiangiogenic treatments, and VEGF antagonists could remodel the TAM phenotype and induce vascular normalization. TAMs also increase the expression of cysteine cathepsins [186] which aid in the recruitment of monocytes to the TME and support cancer cells from various chemotherapeutic drugs [187]. Chemokine conjugates have been designed which could be activated by enzymes that target TAMs in mouse cancer models. These agents were produced by the conjugation of mCCL2-thiol to cathepsin-activatable fluorophores or caged prodrugs [188]. These probes interact with intracellular cysteine cathepsins in macrophages through CCR2-mediated endocytosis. They also release the cytotoxic chemical doxorubicin for the ablation of macrophages or fluorescently marked active cathepsins in macrophages [189]. Angiogenesis is essential for the development and spread of tumors. Anti-angiogenic methods, particularly anti-VEGF therapy, have been FDA approved for the treatment of various malignancies [190]. A high M2-like/M1-like macrophage ratio has been linked to resistance to anti-VEGF antibody (AVA) therapy, and macrophage M2 polarization may be involved in AVA resistance. Therefore, targeting macrophages may be a potentially innovative approach to overcome AVA treatment resistance in ovarian cancer [191]. The interaction between microseminoprotein (MSMP) and CCR2 could promote adaptive resistance to AVA in ovarian cancer models. BET inhibitor (BETi) is a compound that reduces macrophage recruitment, CCR2 [192], and MSMP expression. This compound could also enhance the efficacy of AVA therapy in ovarian cancer by inducing apoptosis in M2-like macrophages and reprogramming them to have an M1-like phenotype. BETi has been shown to overcome resistance to AVA treatment and increase survival in an adaptive resistance model of ovarian cancer (Figure 3) [193].

### 5.2. TAMs and the Immune Checkpoint Blockade (ICB)

Immune checkpoint blockade (ICB) immunotherapy, which activates T cell-mediated type 1 immune response, has become a key treatment approach for cancer [194]. However, myelomonocytic cells, which include macrophages, can contribute to primary (before) and adaptive (after) resistance to ICB through the expression of immunosuppressive molecules, such as checkpoint ligands (CD80, CD86, PDL1, PDL2, etc.) and the poliovirus receptor [195,196]. In vivo studies have demonstrated the association of the expressed PDL1 on immune cells (in the TME), with a response to antibodies against PD1 or PDL1 [197]. Interestingly, the expressed PD1 on TAMs has a contrariwise correlation with their ability to phagocytose tumor cells [198,199]. Additionally, other counter-receptors have been expressed on myelomonocytic cells capable of interacting with regulators expressed by natural killer cells and T cells (negative regulators like VISTA) [200]. This molecule could interact with P-selectin glycoprotein ligand 1 (PSGL1) [201] and function as a T cell checkpoint antagonist. The composition of the microbiome can also influence the response to immunotherapy. For anti-PD1 and anti-CTLA4 [202] treatments, the diversity and frequency of gut flora can shape the infiltration of myeloid cells into tumors [203]. The depletion of macrophages has been shown to enhance the effectiveness of different types of immunotherapeutic methods, comprising vaccination [204] and checkpoint inhibitors [205,206]. Multiple clinical trials are currently underway that combine varied TAM-targeted therapeutic approaches (Table 1).

### 5.3. Targeting of TAMs by Vaccination

Immunomodulatory vaccination is an innovative approach targeting the myeloid cells in TAMs [207]. Anti-regulatory T cells (anti-Tregs) specifically recognize and respond to TAMs by restricting the various immunosuppressive signals they mediate. They could distinguish the HLA-restricted epitopes of arginase, PD-L1, and indoleamine 2,3-dioxygenase (IDO) [208]. Activated anti-Tregs can transform the TME into an immune permissive site. The first IDO vaccinations were conducted in patients with NSCLC (non-small cell lung cancer). A median of 26 months of survival was observed in vaccinated patients, which was significantly longer than the median overall survival of 8 months for the untreated control patients [209]. Currently, there is an industry-sponsored phase II clinical trial underway to test the combination of IDO vaccinations and pembrolizumab as a first-line treatment for non-small cell lung cancer [165]. There is another ongoing phase I trial of PD-L1-based vaccinations in multiple myeloma [210], in which a combination of IDO and PD-L1-specific T cell vaccinations with nivolumab is employed for metastatic melanoma [211]. No observation of toxicity (grade III or IV) indicates good tolerability in vaccinated patients. 

Activation of CD8 and CD4 anti-Tregs is essential in the development of therapeutic immune modulatory vaccines. Current cancer vaccine strategies are based on the induction of cancer-specific CD8 cytotoxic T cells, while the immunosuppressive mode is converted to a pro-inflammatory one via activated anti-Tregs. The pro-inflammatory stimulus can convert not terminally differentiated TAMs into M1 macrophages. CD4 cells are particularly effective cytokine-producing cells. Therefore, the activation of CD4 anti-Tregs may be as essential as the activation of CD8 anti-Tregs in a therapeutic process. This is due to the fact that, unlike other strategies of TAM targeting, the activation of anti-Tregs combines both TAM depletion (through direct killing of T cells) and TAM reprogramming (through the provision of pro-inflammatory cytokines in the immune suppressive microenvironment). These processes are crucial in the rebalancing of the microenvironment and enhancing the effectiveness of checkpoint inhibitors such as T cell-enhancing drugs. In many cancer patients, the infiltration of TAMs into the TME majorly contributes to the limited effect of checkpoint inhibitors. Anti-Tregs activated by therapeutic vaccines can result in T cell gathering, Th1 inflammation induction, and elevation of protein expression like IDO and PD-L1 in cancer and immune cells. This creates more targets that could respond to anti-PD1/PD-L1 immunotherapy. Therefore, immune modulatory vaccines that rebalance the microenvironment could increase the effectiveness of T cell-enhancing drugs like checkpoint inhibitors. Combining these vaccines with checkpoint-blocking antibodies could potentially enhance the number of recovered patients [212].

### 5.4. Macrophage Cell Therapy

The pool of TAMs is continually replenished through the recruitment of circulating monocytes. Macrophage-based cell therapies may have the potential to prevail over this limitation due to the steady influx of mononuclear phagocytes into tumors. These therapies are established based on the modification of mononuclear phagocytes with engineered receptors or the ability of monocytes to deliver nanoparticles or cytokines to the TME. According to an in vivo study, the replenished monocytes with drug-loaded nanoparticles were capable of reaching the tumor cells with higher efficiency compared to free nanoparticles [213]. De Palma et al. explored the possibility of delivering interferon alpha (IFNα) to the tumor cells using macrophages and stimulating an immune-related response [214]. They transduced the Ifna1 gene into hematopoietic progenitors under the control of the Tie2 promoter. These Tie2-expressing monocytes, which had a high affinity for tumors, triumphantly entered tumors, delivered IFNα into the TME, activated immune cells, and inhibited angiogenesis and tumor development [214]. In a similar manner, soft particles called “backpacks” comprising IFNα on their internal side were attached to macrophage surfaces [215]. The study showed that macrophages, which were carrying these backpacks, acquired an M1 phenotype. Moreover, upon intratumoral injection, the phenotype was preserved without being influenced by the immunosuppressive TME. A significant reduction in metastatic tumors was observed in a mouse model treated with macrophages carrying IFNγ backpacks [215]. In a mouse model of sarcoma, pro-metastatic niches were determined by the signature of an immune suppression gene centered on myeloid cells. Genetically engineered myeloid cells expressing interleukin-12 (IL-12) were also seen to trigger a type 1 immune response and reduce primary tumor growth upon adoptive transfer [216]. 

Tanoto et al. reported an engineered macrophage, named “MacTrigger”, capable of inflammation induction in only tumor tissues. According to evidence, the MacTrigger accelerated the release of TNF-α, natural killer cells, and CD8^+^T cells, causing efficient effective anti-tumor effects [217]. The major challenge in creating phagocyte-based cellular therapy is the difficulty of transducing human macrophages. This issue was addressed through the development of various technological scaffolds [218,219]. Human CAR-M cells armed with receptors recognizing CD19, CD22, the carcinoembryonic antigen-related cell adhesion molecule 5 (CEACAM5), CD514, and HER2 [220,221], have been developed to target primary and metastatic tumors, mediate phagocytosis, and stably express M1 functions [79]. Clinical trials are ongoing to examine the potency of CAR-M-based therapies in various tumors [93]. In addition to utilizing polarization for cancer therapy, Aalipour et al. reported a new class of cell-based in vivo sensors as highly sensitive cancer diagnostics which was claimed to be more sensitive than both protein and nucleic acid cancer biomarkers. The engineered immune cells as diagnostic sensors can detect tumors as small as 4 mm [217,222].

### 5.5. Peptides Targeting TAMs

Nanomedicine offers significant advancements in cancer therapy, particularly in terms of improving treatment effectiveness and minimizing side effects. These advantages are achieved through the specific targeting of TAMs. Multiple TAM-specific peptides are currently being examined, including M2pep [168], UNO [169], Melittin [170], RP-182 [171], IL4RPep-1 [223], T4 peptide [173], Pep-20 [174], and CRV [175]. A study by Cieslewicz et al. [224] employed in vitro and in vivo phage peptide display libraries to identify M2pep as a peptide that binds to TAMs. M2pep has demonstrated the ability to reduce TAM levels and enhance the survival of CT26 murine colorectal cancer cells in modeling experiments.

Presently, M2pep is considered a pro-apoptotic peptide and is the primary focus of research on nanocarrier development to deliver CSF-1/CSF-1R inhibitors [168]. Several studies have aimed to improve the stability and targeting capabilities of M2pep [225]. These efforts include modifying M2pep through amino acid substitutions, incorporating decafluorobiphenyl cyclization, and developing a pH-sensitive variant by replacing tyrosine with 3,5-diiodotyrosine [226]. 

### 5.6. Nanoparticles Targeting Macrophages

Recent studies have highlighted that nanoparticles (NPs) targeting macrophages offer two main strategies in the battle against cancer. The first strategy focuses on depleting TAMs, aiming to reduce their tumor-promoting effects. The second strategy emphasizes the reprogramming or re-education of TAMs to unleash their inherent anti-tumor potential [227]. Macrophage depletion can be achieved through various approaches, such as targeting the signaling pathway of colony-stimulating factor 1 (CSF1) and its receptor (CSF1R), which prompts apoptosis in a significant proportion of TAMs [143,228].

In addition, blocking the recruitment of circulating inflammatory monocytes to the tumor site is crucial. This recruitment process relies heavily on the signaling pathway of CC-chemokine ligand 2 (CCL2) and its receptor, CC-chemokine receptor 2 (CCR2). By inhibiting the CCL2-CCR2 signaling pathway, the retention of mononuclear cells in the bone marrow occurs, leading to reduced recruitment to both primary and metastatic tumor sites [229]. Reprogramming TAMs is a promising strategy in cancer treatment, reversing their pro-tumor phenotype to an antitumor one. This approach activates M1 macrophages, promoting the activity of cytotoxic T cells and other effector cells. Small molecules and NP formulations, such as TLR agonists, cytokines, antibodies, and RNAs, are also being explored to achieve macrophage repolarization and inhibit cancer growth.

For instance, Xiao et al. discovered that a micellar nano-drug, through the M2-targeting co-delivery of IKKβ siRNA and STAT6 inhibitor AS1517499, effectively repolarized M2-like TAMs into M1-like TAMs. Furthermore, the nano design was tailored to function in the acidic pH of the TME and minimize off-target effects in normal tissues [230]. Furthermore, it was demonstrated that combining nanoparticle-targeted macrophage strategies with other immunotherapies, such as immune checkpoint blockade, yields significant benefits in cancer treatment. For example, in a study by Rodell et al., it was shown that the in vivo delivery of TLR7/8 agonists to TAMs was effectively achieved through the use of R848-loaded β-cyclodextrin nanoparticles, which led to M1 polarization. When combined with the immune checkpoint inhibitor anti-PD-1, the utilization of these nanoparticles resulted in enhanced response rates to immunotherapy, even in a tumor model that exhibited resistance to anti-PD-1 therapy as a standalone treatment [159]. The challenges in NP-based macrophage-targeting therapies involve optimizing timing for NP delivery, addressing the complexity of macrophage subtypes, and understanding NP-cell interactions. While early clinical trials indicate promise, further work is required to ensure NP safety and efficacy, personalize treatments, and bridge the gap between research and clinical applications [231].

In conclusion, the targeting of tumor-associated macrophages (TAMs) represents a promising avenue for improving cancer therapy. Immune checkpoint blockade (ICB) has revolutionized cancer treatment by activating T cell-mediated immune responses, but TAMs can contribute to resistance through immunosuppressive mechanisms. However, strategies targeting TAMs, such as vaccination and macrophage cell therapy, show potential in overcoming this resistance and enhancing treatment outcomes.

## 6. Macrophage and Hematologic Malignancies

Macrophages residing in the TME of myeloma, lymphoma, or leukemia can provide insights regarding disease progression and the effectiveness of chemotherapy. TAM interactions with other cells in the TME could lead to a pro-tumorigenic environment that includes the promotion of chemo-resistance in cancer cells, stimulation of tumor cell development through the production of growth and matrix remodeling factors, and induction of immunosuppression through influencing the behavior of immune cells [232]. Although the role of TAMs in solid tumors has been under the spotlight in past years, the significance of TAMs in hematologic malignancies has only recently been appreciated, owing to the distinctive and varied microenvironments of these conditions. This review will center on the current preclinical and clinical findings regarding macrophages in hematologic malignancies (Table 2 and Figure 4).

### 6.1. Macrophage Role in Leukemia

Leukemic stem cells (LSCs) share the survival and functional properties of the hematopoietic stem cell (HSC). These cells and the hematopoietic microenvironment can give rise to persistent leukemia, which cannot be completely eradicated. The presented TAMs in the microenvironment of leukemia are called LAMs [274]. Acute lymphoblastic leukemia (ALL), AML, and CLL are three subtypes of leukemia. Recent studies have focused on the significance of TAMs in cancer development.

#### 6.1.1. ALL

The activation of the CXCR4/CXCL12 axis has been shown to block the polarization of TAMs to the M1 phenotype [241]. Plerixafor is an inhibitor of CXCR4, which is reported to improve clinical scores in T-ALL. In a study by Song et al., 97 bone marrow (BM) samples from patients with acute leukemia (26/97 with ALL) were compared to 30 healthy control samples from individuals with iron-deficiency anemia [275]. The count of CD68-, CD163-, and CD206-positive macrophages was notably higher in the leukemic BM samples in comparison with the control group. These cells significantly decreased after therapy in patients who achieved complete remission. Nevertheless, they remained higher than the control group. Considering the CD68 as a pan-macrophage marker, the CD163^+^/CD68^+^ or CD206^+^/CD68^+^ ratio was enhanced in the leukemic BM samples, which further supports M2 polarization. Further, the amount of CD163^+^ cells was an autonomous prognostic issue in these patients. The T-ALL cells co-cultured with M2 macrophages led to significant induction of leukemic cell proliferation through IL-6, growth-related oncogene (GRO)-α, C5a, and TNFα [276]. Hohtari et al. analyzed the immune cell composition in the bone marrow of adult precursor B cell ALL patients. They found an increased amount of M2-like macrophages and myeloid-derived suppressor cells in the BM of ALL patients compared to healthy ones [277]. Various patterns of expressed TAM genes and phenotypes in the BM versus spleen were detected through analysis of multiple lymphoid organs in the Notch1 mouse model with overexpressed T-ALL. It was also demonstrated that splenic TAMs stimulate the growth of T-ALL cells better than bone marrow TAMs [278]. Several studies proposed the efficacy of leukemic cells and TAMs in the TME in the development of ALL. For example, Valencia et al. found that ALL cells release bone morphogenetic protein 4 (BMP4), which can generate M2-like macrophages and induce immunosuppressive dendritic cells. These cells could produce TNFα in low levels and great levels of IL-10, CCL2, and IL-6. [245]. Additionally, a recent report on malignant ALL cells demonstrated that the deletion of stromal interaction molecule 1 (STIM1) and STIM2 restores the pro-inflammatory status of TAMs through IFNγ and reduces the number of infiltrated macrophages [279]. These findings suggest that the interplay between TAMs and leukemic cells and TAMs may be involved in the promotion of tumorigenesis in ALL.

#### 6.1.2. AML

AML is often associated with poor clinical outcomes [280]. One factor that contributes to the high rate of relapse, failure of targeted and traditional treatments, and mortality in AML patients is its resistance to therapy. The mechanisms of resistance in AML treatment are not fully understood. Therefore, finding novel strategies to overcome therapy resistance is essential for successful AML treatment [281]. Previous research has primarily focused on the mechanisms of therapy resistance that are inherent to leukemic cells, such as TP53 mutations. These studies have not extensively examined the mechanisms of acquired resistance that occur through exterior processes [282]. However, recent evidence suggested that the interplay between leukemic cells and other cells in the bone marrow microenvironment (BMME) can lead to acquired therapy resistance in AML.

Recently, Moore et al. found that bone marrow macrophages could decrease the growth of AML in animals through a process called LC3-associated phagocytosis. This process involves the phagocytosis of dying and dead leukemic cells, which includes the mitochondria within the leukemic blasts. These functions could activate the stimulator of IFN genes (STING) and lead to the production of inflammatory signals that enhance phagocytosis and inhibit the expansion of leukemic cells [283,284]. High levels of CD16/CD32 and CD64 are expressed in the spleen and bone marrow macrophages as Fc-activating receptors, which lead to the inhibition of AML through phagocytosis [285]. CD200 is a protein that is overexpressed in AML stem cells (LSCs). This protein can attenuate the response of macrophages to AML [285]. Moreover, treatment with an anti-CD200 antibody can specifically facilitate the phagocytosis of CD200^+^ AML cells by macrophages through a process called antibody-dependent cell phagocytosis (ADCP) [286]. AML cells that had mutated DNA (cytosine-5)-methyltransferase 3A (DNMT3A) were found to inhibit the polarization of M1 macrophages and resist their killing effect in the laboratory and animal models. In animals with xenografts (transplants of human tumors into mice), the experimental group had significantly larger tumor volumes and a higher proportion of M2 macrophages compared to the control group [287]. Interleukin 4 (IL4) has a powerful anti-leukemic effect in mice by promoting the phagocytosis of AML cells by macrophages. IL4 stimulation leads to the upregulation of CD47 in a STAT6-dependent manner. Moreover, the combination of IL4 stimulation with CD47 blockade further enhances the phagocytosis of AML cells by macrophages [288]. Chenodeoxycholic acid (CDCA) [289] is a type of bile acid, which can inhibit the polarization of M2 macrophages. These cells may have a synergistic effect on reducing the progression of AML. A potential target for chimeric antigen receptor T cell (CAR-T) therapy of AML is the C-type lectin domain family 12 member A (CLEC12A). Its expression level is closely linked to treatment response and patient survival outcomes. The expression of CLEC12A is positively correlated with the infiltration of type 2 macrophages and monocytes [290]. Peritoneal resident macrophages in AML-AF9-induced mice had an M2-like phenotype, which can contribute to cancer progression [291].

Studies determining the function of macrophages (Mφs) in AML have been limited by challenges in accurately distinguishing non-malignant from malignant or AML-associated Mφs. Conventional methods such as immunohistochemistry and flow cytometry have been routinely used for AML patients to determine M2-like Mφs/monocytes in the bone marrow or spleen based on myeloid markers such as CD163 and CD206. Nevertheless, these myeloid markers are also expressed on Mφs, non-malignant monocytes, and AML-associated Mφs. Recently, the detection of mutations and transcript expression discrepancy has been facilitated using single-cell RNA sequencing and genetic profiling. These methods allow for the specific characterization of malignant and non-malignant Mφs within the BMME of AML [292]. Despite improvements in the identification of TAMs/M2-like Mφs, our knowledge of the biology of Mφ in AML has just started to develop. Significant questions remain about the different Mφ groups within the BMME and their role in disease development. Given new technologies like CO-Detection by indEXing (CODEX) [293], the interplay between AML blasts and the surrounding BMME could be visualized. Single-cell sequencing technologies like MacSpectrum employ single-cell RNA sequencing data and can distinguish distinct macrophages derived from bone marrow and adipose tissue. These methods are also contributing to our knowledge of the complex function of macrophages in different diseases, like AML. It is important to comprehensively characterize tissue-resident Mφs and LAMs in the BMME to explore molecular differences for the precise targeting of LAMs. This will be crucial in generating novel Mφ targeting strategies with improved efficacy and declined toxicity. Possible combinations for the treatment of Mφ-mediated therapy resistance, such as selumetinib and/or AZD5991 or CYC065, could be considered as new therapeutic approaches to prevail Mφ- and MCL-1-driven therapy resistance in AML. In light of these facts, the future holds great promise for the development of unprecedented therapies targeting Mφ-mediated immunomodulation in AML [294].

#### 6.1.3. CLL

CLL is a common and frequent type of leukemia in the elderly population. CLL and its related condition, known as small lymphocytic lymphoma (SLL), are recognized as belonging to the category of mature B cell neoplasms by the World Health Organization classification [295]. CLL can range from mild to aggressive in terms of its symptoms, and its treatment can range from a watchful waiting approach to immediate treatment [221]. TAMs, also known as NLCs (Nurse-like cells) in CLL [296], are a part of the TME and resemble M2-polarized macrophages. The level of TAM infiltration has been correlated with a poor prognosis, but this has not yet been proven for CLL [297]. Studies have shown that isolated CLL cells die in vitro, but when co-cultured with NLCs, they can proliferate. This observation suggests that the key to a cure for CLL may lie in the features of the TME and tumor cells. A better mechanistic grasp of the TME could lead to the development of efficient cancer therapies that target its modulation. These therapies could ring about personalized cancer treatments with better tolerance and fewer side effects [298].

The TNFR (tumor necrosis factor receptor) ligand, known as APRIL, has a remarkable role in the proliferation of CLL cells. However, the exact mechanism has not been revealed. Van Attekum et al. examined the role of APRIL in various aspects of CLL biology using a co-culture system with APRIL overexpression, recombinant APRIL, and APRIL reporter cells [299]. They found that APRIL had no effects on the survival of CLL cells in these systems and did not enhance the activation of NF-κB or affect CLL proliferation in single or combined stimuli. Additionally, the survival effect of macrophages on CLL cells was not affected by the APRIL decoy receptor transmembrane activator and CAML interactor-Fc [300]. These results suggest that the direct role of APRIL in CLL cell survival may have been overestimated due to the use of high levels of recombinant APRIL. Nurse-like cells (NLCs), also known as CLL-specific TAMs expressing CD68 and CD163 [301], have been shown to protect the CLL B cells from apoptosis through stromal cell-derived factor-1 [296]. NLC differentiation includes significant DNA methylation changes, which are MEK pathway dependent. MEK inhibitors reduce NLC numbers in vitro and may decrease the number of splenic monocytes/macrophages, which are mainly the M2-like population. The M2-like phenotype was observed in NLCs from high-viability CLL cultures. These cells can attract and facilitate contact with cancer cells, which has been linked to their protective function. In contrast, NLCs from low-viability CLL cell cultures show an M1-like phenotype and do not attract CLL cells. The addition of IL-10 to the culture can induce an M2-like phenotype in NLCs and increase CLL cell viability. On the other hand, TNF can depolarize protective M2-like NLCs into non-protective M1-like NLCs. IL-10 can repolarize TNF-depolarized NLCs and restore their protective effect on CLL cells [302].

### 6.2. Macrophage Role in Lymphoma

The progression of lymphoma could be supported by macrophages both in classic Hodgkin’s lymphoma (CHL) and non-Hodgkin’s lymphoma (NHL) (Figure 4).

#### 6.2.1. Hodgkin Lymphoma

Studies have shown that the presence of CD163^+^ macrophages in tumor tissue is related to poor survival in patients with classical Hodgkin lymphoma (CHL) [303]. Additionally, a lower number of TAMs in CHL samples was correlated with a higher progression-free survival rate [304]. The lower level of M2 macrophages has been linked to a complete response and better survival [305]. The expressions of both TAM markers, CD68, and CD163 [306] are essential predictors of complete remission in CHL patients [307]. A high ratio of LAMs to Hodgkin–Reed–Sternberg cells at diagnosis is associated with a higher risk of CHL progression or death [308].

#### 6.2.2. Non-Hodgkin Lymphoma

The number of CD68^+^ and CD163^+^ macrophages significantly increases in all three grades of follicular lymphoma [309]. A high PD-1 expression on TAMs in the T cells of non-Hodgkin lymphoma may predict a poor prognosis. It enhances the pro-tumor effects of the TME and inhibits the polarization of M1 macrophages and phagocytosis [310]. A high M2 TAM content at diagnosis, particularly in combination with an international prognostic index, may be a factor in the identification of diffused large B cell lymphoma patients [311]. GM-CSF amplifies the inhibitory effect of CHOP chemotherapy on DLBCL progression by promoting the polarization of M1 macrophages [312]. Enhanced M2 macrophage activation and lipid metabolism have been observed in the immunosuppressive tumor microenvironment of non-MYC/BCL2 double express or DLBCL [313]. The LXRα-related signaling pathways and functions are connected to M1 polarization and may increase the immune reactivity of macrophages in DLBCL [314].

### 6.3. Macrophage Role in Multiple Myeloma

Macrophages are a type of immune cells that are prevalent in the bone marrow of individuals with multiple myeloma (MM) and can support the proliferation, induce drug resistance, and contribute to the formation of an immunosuppressive environment. Beider et al. demonstrated that the interactions between macrophages and MM tumor cells result in the polarization of macrophages toward an M2 phenotype [315]. This process increases the production of CXCL13 and activates osteoclasts, which have the ability to resorb bone and promote MM progression. IL-32γ can promote drug resistance in MM through macrophages and modify macrophages towards an M2 phenotype [316]. Increased TAMs in MM patients can stop the functions of cytotoxic T lymphocytes (through the PD-1/PD-L1 pathway) and contribute to the evasion of the immune system by myeloma cells [317]. Exosomes, derived from MM containing IL-32γ, can increase the expression of PD-L1 by macrophages and lead to immune evasion. The PFKFB3-JAK1(6-phosphofructo-2-kinase/fructose-2,6-biphosphatase 3-Janus kinase 1) axis may also play a role in the expression of PD-L1 by macrophages [318]. Gao et al. found that daratumumab (DAR) has a significant anti-tumor effect on MM in mice through its interaction with macrophages via Fc-FcγR. DAR induces the activation of macrophages in mice and results in the phagocytosis of cancer cells through the Fc-FcγR interaction [319]. RGS12 (regulator of G-protein signaling 12) can inhibit the progression and metastasis of MM by the induction of M1 macrophage polarization and activation in the bone marrow microenvironment (BMME) (Figure 4) [320].

## 7. Challenges in TAM-Based Therapeutics (in Solid or Hematologic Tumors)

The reduction in negative side effects with TAM-based strategies is an ongoing challenge. Due to the complicated functions of macrophages in maintaining homeostasis, TAM depletion could increase the risk of infections or disorganize tissue-resident cells and prevent them from performing their usual functions. Therefore, discovering TAM-specific molecules or markers that are mainly created through metastasis-associated macrophages (MAMs) and/or activated M2 (AAMs) will enable therapeutic approaches to specifically target tumor cells without affecting the normal function of other immune cells which are tissue-resident [321]. The potency of wound healing and phagocytosis in non-tumor tissues should be preserved as a goal of the techniques which target macrophage reprogramming.

## 8. Conclusions

Macrophages are endowed with the ability to adapt and change their function based on external stimuli. They are prevalent in the TME and have an indispensable role in cancer progression. Various efforts have been made to alter the behavior of TAMs and inhibit their functions in the promotion of tumor growth. However, the ability of macrophages to travel to both primary tumors and metastatic sites presents an opportunity for their application as a means for the delivery of cellular therapies to cancer cells. As antigen-presenting cells, macrophages link innate immune responses with adaptive immunity. The development of gene engineering techniques, such as the use of Vpx-LV and Ad5f35 as vectors for modification of primary human macrophages, has opened the possibility of redirecting the function of these cells against tumors through synthetic biology. In addition, immune modulatory vaccines, which target TAMs in the TME, have emerged as an alternative to traditional antibodies or small molecule inhibitors and have shown promise in preclinical and clinical trials.

## Figures and Tables

**Figure 1 cancers-15-03722-f001:**
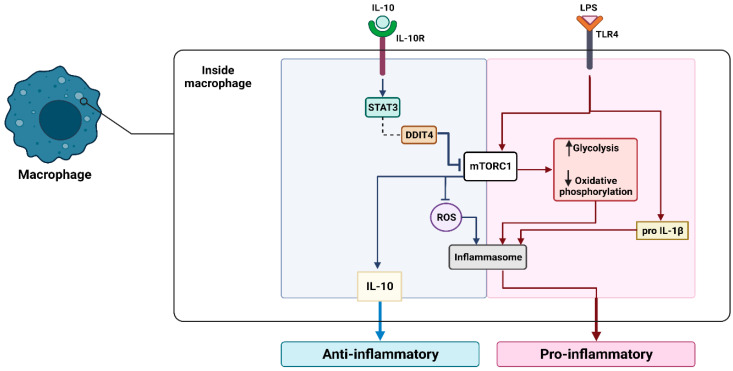
The released DAMPS due to cell death within the tumors can activate macrophages and result in immunosuppression via the elevated production of IL-10, which suppresses the expression of pro-inflammatory cytokines and promotes the formation of Tregs. Moreover, macrophages can contribute to the secretion of IL-6 and TNFα, leading to cancerous cell survival. In addition, TNFα activates the NF-κB pathway upon binding to the TNFR1/2 receptor, controls the expression of VEGF, IL-6, and the stimulation of neo-angiogenesis. IL-6 could then promote cell proliferation and differentiation via the JAK/STAT3 pathway. Phagocytosis, the propagation of TNFα, and interleukin-1 beta (IL1β) can induce macrophage recruitment.

**Figure 2 cancers-15-03722-f002:**
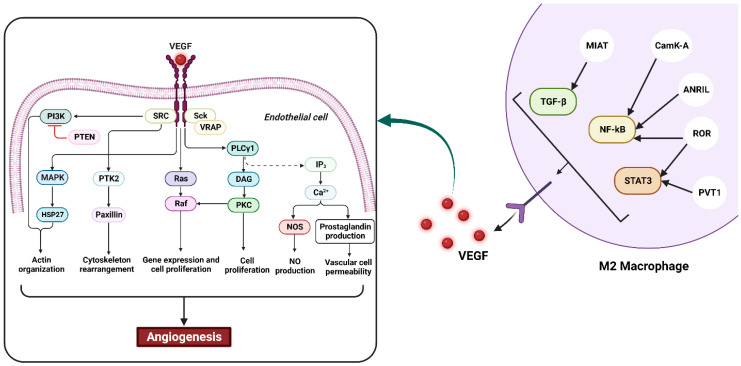
Role of macrophages in angiogenesis. Macrophages are considered to play a critical role in inflammatory and tumor angiogenesis due to their widespread existence in healthy and inflamed tissues, their ability to become activated in response to certain stimuli, and their production of various secretory agents. Alternatively activated macrophages could release proteases, various growth factors, and monokines, including TGF-alpha, bFGF, GM-CSF, VEGF/VPF, IGF-I, PDGF, TGF-beta, IL-1, IL-6, substance P, IL-8, TNF-alpha, interferons, prostaglandins, and thrombospondin 1. These factors can influence each phase of angiogenesis, such as the induction of endothelial cell migration or proliferation, changes to the local extracellular matrix, and inhibition of vascular growth via the formation of differentiated capillaries.

**Figure 3 cancers-15-03722-f003:**
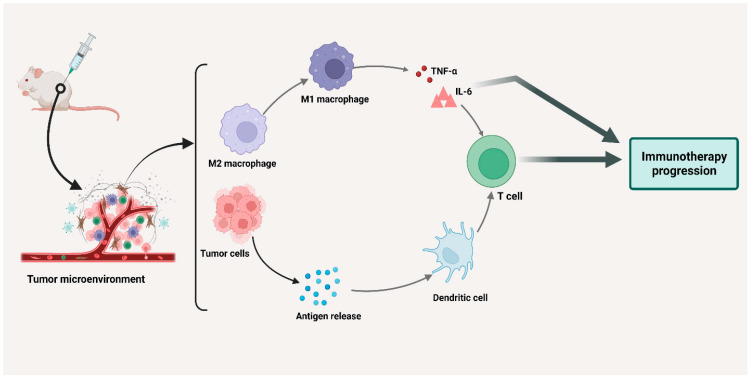
TAMs in immunotherapy.

**Figure 4 cancers-15-03722-f004:**
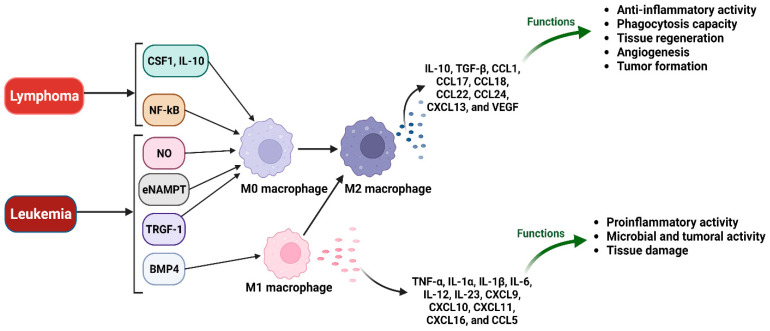
Macrophage and hematologic cancers. Interactions between macrophages residing in the TME and tumor cells finally result in polarization toward M2 macrophages which can provide insights regarding disease progression. Various small peptides belonging to the CXC chemokine family (including chemokine (C-X-C motif) ligand 10 (CXCL 10), CXCL11, and CXCL 13 and 16), the CC chemokine family (including chemokine ligand 1 (CCL1), CCL5, 17, 18, 22, and CCL24) are produced during this process. In addition, transforming growth factor beta (TGF-β), vascular endothelial growth factor (VEGF), tumor necrosis factor alpha (TNFα), interleukin-1α (IL-1α), 1 β, 10, 12, and IL-23 can also contribute to different functions, as shown.

**Table 1 cancers-15-03722-t001:** Different strategies in targeting macrophages for treatment.

Strategy	Pathway	Target	Agent/Drug(s)	Type of Tumor	Result(s)	Ref. or Trial No.
Conventional	TAM depletion	CSF-1R	BLZ945	Solid tumors	Enhancing the level of CD8^+^ cytotoxic T cells leading to the prevention of tumor growth	[94]
		CSF-1R	PLX3397 (Pexidartinib)	Sarcoma, breast cancer, prostate cancer, and solid tumors	Infiltration of T cells in the TME	[95,96]
		IL10/VEGF/TGFβ	Zoledronic acid	Breast cancer	Infiltration of CD8^+^ T cells and improving immune responses	[97]
		Pan-macrophages	Trabectedin	Soft tissue sarcomas and recurrent ovarian cancer	Causing selective cytotoxicity to TAM populations by triggering the extrinsic TRAIL apoptotic pathway	[98,99,100,101]
		Pan-macrophages	Lurbinectedin	Ovarian cancer and solid tumors	Eliminating tumor cells directly through the TRAIL-dependent apoptosis pathway and a reduction in angiogenesis	[102,103,104,105]
		CSF1-R	ARRY-382	Solid tumors	Not determined	[106]
		CSF1-R	AMG820	CRC and solid tumors	Not determined	[106,107]
		CSF1-R	Emactuzumab	Solid tumors	Inhibiting the activation of CSF1R	[108]
		MMP-2	Doxorubicin	Melanoma, Breast cancer	Reduction in Treg infiltration to the TME	[109]
		MMP-2	Clodronate	Bone metastatic cancers	Suppressing tumor growth and angiogenesis	[110]
		**Target**	**Drug(s)**	**Type of tumor**	**Result(s)**	**Ref. or Trial no.**
	Blocking recruitment	CCL2	Carlumab	Prostate cancer	Blocking CCL2 signaling leading to tumor growth prevention	[111]
		CCL2	CNTO 888	Solid tumors	CCL2 inhibition	[112]
		CCR2	Propagermanium	Breast cancer	CCL2 inhibition	[113]
		CCR2	PF-04136309	Pancreatic cancer	Reducing the circulatory CCR2^+^ monocytes and an increase in bone marrow CCR2^+^ monocytes	[114]
		CCR2	BMS-813160	CRC and pancreatic cancer	Inhibition of inflammatory monocytes and macrophages migration	[115]
		CCR5	Leronlimab	Breast cancer	Inhibition of tumor development, adhesion, and invasion	[116,117]
		CCL5	Maraviroc	Metastatic colorectal cancer	Inducing M1 like TAMs polarization, which mediated antitumor responses	[118]
		CCR5	Vicriviroc	Metastatic colorectal cancer	CCR5 inhibition	[119]
		CCR5	TAK-779	Colorectal cancer	CCR5 inhibition	[120]
		CCR5	Anibamine	Ovarian cancer cells	CCR5 inhibition	[121]
		CCR5	GSK706769	Colorectal cancer	CCR5 inhibition	[122]
		CX3CL1	DC101/anti-Ly6G antibody	Colon cancer	Inhibition of macrophage recruitment in the TME	[123]
		CSFR-1R	Pexiddartinib (PLX-3397)	Tenosynovial giant cell tumors and other solid tumors	CSF-1R inhibition	[124]
		CSFR-1R	Chiauranib (CS2164)	Solid tumors	Inhibition of CSF-1R and angiogenesis-related kinases (VEGFR, PDGFRa, and c-Kit)	[125]
		CSFR-1R	RG7155	Solid tumors	Reducing CSF-1R, CD163, TAMs, and peripheral blood CCR2Monocyte	[126]
		CSFR-1R	Cabiralizumab	Solid tumors	CSF-1R inhibition	[127]
		CSFR-1R	AZD7507	Pancreatic cancer	CSF-1R inhibition	[128]
		CX3CL1	JMS-17-2	Breast cancer cells	Metastatic seeding and colonization of breast cancer cells	[129]
	TAM repolarization	**Target**	**Drug(s)**	**Type of tumor**	**Result(s)**	**Ref. or Trial no.**
		CD40	CP-870 and 893	Melanoma, pancreatic cancer, and solid tumors	Stimulation of adaptive immune responses and M1 macrophage activation and cancer cell apoptosis	[130,131,132,133] NCT02225002
		CD40	APX005M (Sotigalimab)	Melanoma and pancreatic cancer	Inducing T cell-dependent tumor regression and improving survival	[134] NCT02706353
		CD40	CDX-1140	Melanoma and breast cancer	Activating DCs and B cells and leading to NFkB stimulation in CD40-expressing cells	[135] NCT04616248
		CD40	SEA-CD40	Solid and hematological tumors	Binding with increased affinity to FcγRIIIa resulting in an enhanced effector function and CD40 agonism	[136] NCT02376699
		CD47/SIRPα	TTI-621	Hematological malignancies	Increasing cancer cells phagocytosis by macrophages and antigen presentation which activate T cells	[137,138]
		CD47/SIRPα	Magrolimab	Solid and hematological tumors	CD47 inhibition	[139]
		CD47/SIRPα	Hu5F9-G4	Solid tumors	CD47 inhibition	[140]
		CD47/SIRPα	IBI188	Solid tumors	CD47 inhibition	[141]
		CD47/SIRPα	ZL1201	Solid tumors	CD47 inhibition	[141]
		CD47/SIRPα	BI 765063	Solid tumors	SIRPα inhibitors	[141]
		CD47/SIRPα	CC-9525	Solid tumors	SIRPα inhibitors	[141]
		CD47/SIRPα	ChiLob7/4	Various tumors	CD40 agonists	[142]
		CSFR-1R	BLZ945	Solid tumors	Reducing M2 associated gene expression (Adm, Arg, Mrc1, and F13a1) in TAMs	[143]
		CSFR-1R	PLX3397	Glioma	TAM repolarization and consequent tumor suppression	[144]
		-	Membrane-coated Fe_3_O_4_ nanoparticle	Melanoma	Re-educating M2-macrophages to M1, decreasing cancer’s metabolic function, and induction of immunologic cell death	[145]
		TLR7/8 agonist	Resiquimod	Melanoma	M2 repolarization into M1 and elevating the level of antibody-dependent cellular phagocytosis	[146,147]
		TLR7/8 agonist	TransCon	Solid tumors	Enhancing tumor growth inhibition	[148]
		TLR3 agonist	BO-112	CRC, gastric cancer, and melanoma	Re-education of M2 macrophages towards M1 and inhibition of tumor growth	[149,150] NCT04508140
		TLR9 agonist	CMP-001 (vidutolimod)	Melanoma	Upregulating IFN-responsive genes	[151]
		TLR7 agonist	SHR2150	Solid tumors	Immunostimulating and antineoplastic activities	NCT04588324
		PI3K	IPI-549	Solid tumors	Enhancing NFκB activation preventing tumor growth and elevated cytotoxic T cell activity	[152,153]
		MARCO	MARCO mAb	Melanoma, colon, and breast cancer	Upregulating the level of regulatory T cells and anti-inflammatory cytokine IL-37, decreasing tumor growth	[154,155]
		Ferumoxytol nanoparticles	Carboxy-dextran coated super paramagnetic ironoxide nanoparticles (SPIONs)	-	Inducing TAMs phenotypic shift towards tumor-suppressive phenotype and activation of the MAPK pathway	[156]
		Polystyrene nanoparticles functionalized with carboxyl or amino groups	poly(styrene-co-maleic anhydride) (PSMA) nanoparticles conjugated with polymer poly [2-methoxy-5-(2-ethylhexyloxy)-1,4-phenylenevinylene, PPV]	-	Impairing CD163 and CD200R expression and IL-10 production in M2 macrophages	[157]
		Cationic polymers	Cationic dextran and polyethyleneimine (PEI)	Sarcoma	Changing TAM phenotype via TLR4 signaling	[158]
		TLR7/8 agonist	R848 (TLR7/8 agonist)-loaded β-cyclodextrin nanoparticles	Colorectal cancer	Re-education of M2-macrophages to M1, enhancing response rates to immunotherapy when combined with the immune checkpoint inhibitor anti-PD-1	[159]
Macrophage cell therapy (CAR-M)		**Target**	**Drug(s)**	**Type of tumor**	**Result(s)**	**Ref. or Trial No.**
		HER2	CT-0508	HER2^+^ solid tumors	Trafficking into the tumor, phagocytosing and killing cancer cells	NCT04660929
		-	TEMFERON	Glioblastoma	Temferon is well tolerated by patients	[160]
Immune checkpoint blockade (ICB) immunotherapy		PDL-1	-	NSCLC and other tumors	Enhancing the cytotoxic function of T cells	[161]
		VISTA	-	Myeloid cells	Interacting with P-selectin glycoprotein ligand 1 (PSGL1), functioning as a T cell checkpoint inhibitory ligand	[162]
		TIM4	-	Renal cell carcinoma (RCC)	Suppressing CD8^+^ T cell responses, blocking TIM4 with antibodies, and enhancing the efficacy of ICB at these sites	[163]
Vaccine		-	Exosomes derived from M1- but not M2-polarized macrophages	-	Boosting the antitumor vaccine by eliciting a release of Th1 cytokines and a stronger antigen-specific cytotoxic T cell response	[164]
		Indoleamine 2,3-dioxygenase (IDO)	-	Non-small cell lung cancer	Eliciting CD8^+^ and CD4^+^ T cell-mediation	[165]
		Sipuleucel-T	-	Prostate cancer	Inducing antigen-specific T cells with a fusion protein combining a targeting tumor antigen prostate acid phosphatase with GM-CSF, prolonging the survival of patients in a few clinical trials	[166]
		STING agonist	-	Multiple established tumors	-	[167]
Apoptotic peptides		CD206	M2pep	Colon cancer	Murine TAMs (CD45^+^F4/80^+^CD301^+^)	[168]
		CD206	UNO	Solid tumors	CD206 TAMs binding to CD206^+^ (M2) macrophages	[169]
		CD206	Melittin	Solid tumors	CD206 TAMs	[170]
		CD206	RP-182	Solid tumors	CD206 TAMs	[171]
		IL-4R	IL4RPep-1	Breast cancer	IL-4R-expressing macrophages	[172]
		Tyrosine-protein kinase receptor (Tie2)	T4 Peptide	Breast cancer	Tyrosine-protein kinase receptor (Tie2) expressing macrophages (TEMs)	[173]
		CD-47	Pep-20	Wilde range	CD-47	[174]
		Retinoid X receptor beta	CRV	Breast tumors	TAMs retinoid X receptor beta, a receptor found to be expressed predominantly by TAMs	[175]

**Table 2 cancers-15-03722-t002:** Macrophage and hematologic malignancies.

Hematological Malignancies	Disease	Drug/Agent	Study Status/Stage	Mechanism/Observation(s)	References
Leukemia	CLL	Trabectedin	Preclinical evaluation	Antiangiogenic and macrophage killing due to CCL2-CCR2 signaling axis inhibition	[233]
		CSF-1R signaling inhibitor	Preclinical evaluation	CSF-1R signaling inhibition	[234]
		GW-2580, ARRY-382	Preclinical evaluation	CSF-1R signaling inhibition	[235]
		Lenalidomide	Preclinical evaluation	Modifying the TME via promotion of T and NK cell functions and downregulating anti-inflammatory and proangiogenic cytokines	[236]
		TG-1801 (NI-1701)	Clinical/phase I	CD47/SIRPα-targeted bispecific antibodies	[237]
		SRF231	Clinical/phase IA/IB	CD47 inhibition	[238]
	ALL	BLZ-945	Preclinical evaluation	CSF-1R signaling inhibition	[239]
		PLX3397	Preclinical evaluation	CSF-1R signaling inhibition	[240]
		CXCR4 inhibitor plerixafor	Preclinical evaluation	CXCR4/CXCL12 axis inhibition	[241]
		Preemptive IFN-α	Preclinical evaluation	TAM reprogramming	[242]
		Anti-CD47 mAb	Preclinical evaluation	Enabling phagocytosis of tumor cells by TAM	[243]
		CD204-positive TAM	Preclinical evaluation	CD204-positive TAM was associated with malignant cells proliferation, measured according to the Ki-67 labeling index	[244]
		BMP4	Preclinical evaluation	Inducing immunosuppressive dendritic cells and favoring the generation of M2-like macrophages with pro-tumoral features	[245]
		Exposure to myeloid differentiation promoting cytokines	Preclinical evaluation	B-ALL blasts reprogramming into Macrophage	[246]
	AML	Artesunate	Preclinical evaluation	TAM reprogramming JAK2/STAT3 Downregulation	[247]
		Pacritinib	Preclinical evaluation	CSF1R inhibition with a JAK2/FLT3 inhibitor, depletion of TAMs, and, consequently, inhibited leukemic cell survival	[248]
		Hu5F9-G4	Clinical/phase I	Anti-CD47 led to hemoglobin decline and increased transfusion requirements	[249]
	AML	Hu5F9-G4 + Atezolizumab	Clinical/phase I	CD47 inhibition	[250]
		ALX148	Clinical/phase I/II	SIRPα fusion protein that blocks CD47	[251]
		AK117	Clinical/phase I/II	CD47 inhibition	[252]
		IBI188	Clinical/phase IB	CD47 inhibition	[253]
	AML/MDS	Hu5F9-G4	Clinical/phase II	CD47 inhibition	[254]
		TJC4	Clinical/phase IB	CD47 inhibition	[255]
		IMM-01	Clinical/phase I/II	SIRPα fusion protein that blocks CD47	[256]
		CC-90002	Clinical/phase I	CD47 inhibition	[257]
		DSP107	Clinical/phase II	CD47/SIRPα-targeted bispecific antibodies	[258]
	TP53 Mutant AML	Hu5F9-G4	Clinical/phage III	CD47 inhibition	[259]
Lymphoma	Hodgkin lymphoma (HL)	PLX3397	Clinical/phase II	Highly selective inhibitor of CSF1R and Kit receptor tyrosine kinases	[260]
		Brentuximab Vedotin	Clinical/phase IV	An anti-CD30 antibody–drug conjugate	[261]
		Mocetinostat	Clinical/phase II	An oral isotype-selective histone deacetylase inhibitor	[262]
	Non-Hodgkin lymphoma (NHL)	Hu5F9-G4 + Rituximab	Clinical/phase II	CD47 inhibition	[263]
		Hu5F9-G4 + Rituximab + Acalabrutinib	Clinical/phase I	CD47 inhibition	[264]
		IMM0306	Clinical/phase I	CD47/SIRPα-targeted bispecific antibodies	[265]
		ALX148	Clinical/phase I/II	Inhibiting CD47-SIRPα checkpoint	[266]
		Gentulizumab	Clinical/phase I	CD47 inhibition	[267]
		Anti-CD47 mAb Hu5F9-G4	Clinical/phase II	Enabling phagocytosis of tumor cells by TAM	[263]
		Dacetuzumab	Clinical phase II	Anti-CD40 mAb	[268]
Myeloma	MM	Trabectedin	Trabectedin	Antiangiogenic and macrophage killing due to CCL2-CCR2 signaling axis	
		TTI-621	Phase Ib	SIRPα-IgG1 Fc fusion protein inhibiting CD47-SIRPα Checkpoint	[137]
		TTI-622	Phase Ia/Ib	SIRPα-IgG1 Fc fusion protein inhibiting CD47-SIRPα Checkpoint	[269]
		AO-176	Phase I/II	Humanized IgG2 anti-CD47 mAb inhibiting CD47-SIRPα Checkpoint	[270]
		SRF231	Phase Ia/Ib	Fully human anti-CD47 mAb inhibiting CD47-SIRPα Checkpoint	[271]
		BI-505	Phase I	Fully human anti-ICAM-1 mAb overcoming immunosuppression	[272]
		Dacetuzumab	Clinical phase II	Anti-CD40 mAb	[268]
		IBI-322	Clinical/phase I	CD47/SIRPα-targeted bispecific antibodies	[273]

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
