# Peer review of "Macrophage-Based Therapeutic Strategies in Hematologic Malignancies"

_cancers, 2023, doi:10.3390/cancers15143722_

Round 1

Reviewer 1 Report

Overall, this is a clear, concise, and well-written review on the role of macrophages in hematologic malignancies. In this review, Khalili et. al has done an excellent job of covering various aspects of macrophages including their role in cancer cells, their role in inflammation, and their potential as therapeutic targets as single or combination agents. 

The article is well-prepared and good for publication in the Journal.

Author Response

Response: Thank you for your consideration and time.

Reviewer 2 Report

The complicated function of tumor-associated macrophages in tumor growth and their influence on the tumor microenvironment are highlighted by Khalili et al. in this comprehensive review.

The review specifically focuses on hematologic tumors, providing insights into TAMs as a therapeutic target in this context, but also provides information about TAMs involvement in solid malignant tumors.

The review emphasizes the ongoing research to understand the precise mechanisms by which macrophages contribute to tumor growth and the potential therapeutic approaches targeting TAMs. The combination of TAM depletion and immune modulatory vaccination is discussed as a strategy to reverse tumor resistance and enhance cancer response to T-cell-based treatments.

The article is well writen. The bibliography that was assessed is quite substantial.

I would like to congratulate the authors for their diligent work.

Author Response

Thank you very much for your prompt attention to our manuscript. We are very pleased to receive your positive message and we appreciate the opportunity to be able to respond to your comments.

We hope that you will be satisfied by our responses and that you will now be able to accept the paper for publication.

We wish to express our appreciation to the Reviewers for their insightful comments, which have helped us significantly to improve our manuscript. According to the suggestions, we have thoroughly revised our manuscript and its final version is enclosed. Point-by-point responses to the comments are listed below.

Response: Thank you for your consideration and time.

Reviewer 3 Report

In this review, the authors summarized the significant roles of tumor-associated macrophages (TAMs) in various stages of cancer development. Although the structure of the paper is well organized and the topic is interesting and important, some comments need to be addressed to improve the quality:

1.     More professional words should be used, such as rephrasing “immune-suppressive” to “immunosuppressive” for clarity.

2.     The words need to be softened, such as using “potential therapeutic approaches” instead of “prominent therapeutic approaches” to avoid overstating the case.

3.     There are some typing mistakes as well and the author are advised to carefully proof-read the text, for example, the myeloma line in the last table, atiangiogenic should be corrected as antiangiogenic.

In this review, the authors summarized the significant roles of tumor-associated macrophages (TAMs) in various stages of cancer development. Although the structure of the paper is well organized and the topic is interesting and important, some comments need to be addressed to improve the quality:

1.     The layout of this manuscript must be rearranged according to the requirement of MDPI journal before resubmission. 

2.     More professional words should be used, such as rephrasing “immune-suppressive” to “immunosuppressive” for clarity.

3.     The words need to be softened, such as using “potential therapeutic approaches” instead of “prominent therapeutic approaches” to avoid overstating the case.

4.     There are some typing mistakes as well and the author are advised to carefully proof-read the text, for example, the myeloma line in the last table, atiangiogenic should be corrected as antiangiogenic.

Author Response

Thank you very much for your prompt attention to our manuscript. We are very pleased to receive your positive message and we appreciate the opportunity to be able to respond to your comments.

We hope that you will be satisfied by our responses and that you will now be able to accept the paper for publication.

We wish to express our appreciation to the Reviewers for their insightful comments, which have helped us significantly to improve our manuscript. According to the suggestions, we have thoroughly revised our manuscript and its final version is enclosed. Point-by-point responses to the comments are listed below.

Reviewer 3:

  1. The layout of this manuscript must be rearranged according to the requirement of MDPI journal before resubmission. 

Response: According to the reviewer’s comment. The whole text was rearranged according to the requirement of MDPI journal. The manuscript comprise the front matter, literature review sections and the back matter. The Figures, and Tables were also corrected.

  1. More professional words should be used, such as rephrasing “immune-suppressive” to “immunosuppressive” for clarity.

Response: Considering the reviewer’s suggestion, some words were rephrased for clarity which can be detected in track changed/ highlighted format.

  1. The words need to be softened, such as using “potential therapeutic approaches” instead of “prominent therapeutic approaches” to avoid overstating the case.

Response: Thank you for your consideration; it was corrected and highlighted.

  1. There are some typing mistakes as well and the author are advised to carefully proof-read the text, for example, the myeloma line in the last table, atiangiogenic should be corrected as antiangiogenic.

Response: According to the reviewer’s comment, the whole text was carefully proof-read and typing mistakes were corrected.

Reviewer 4 Report

This review is interesting, but some sections or sentences should be added. Major revision should be made before re-submission. This manuscript would be re-considered only when the authors respond all comments.

1.

The mechanism or expression protein in macrophage polarization should be indicated.

2. Macrophage cell therapy

Recently macrophage medicine utilizing polarization for cancer therapy (https://doi.org/10.1016/j.jconrel.2023.04.010) or diagnosis (Nature Biotechnology volume 37pages531–539 (2019)), respectively named as MacTrigger and Macrophage sensor. Utilizing macrophage polarization is one of the most important factors in using macrophage. These methods are capable of effect switching under TME. The authors should describe these papers.

3.

The authors should mention the nanoparticles for TAM target.

Not so bad.

Author Response

Thank you very much for your prompt attention to our manuscript. We are very pleased to receive your positive message and we appreciate the opportunity to be able to respond to your comments.

We hope that you will be satisfied by our responses and that you will now be able to accept the paper for publication.

We wish to express our appreciation to the Reviewers for their insightful comments, which have helped us significantly to improve our manuscript. According to the suggestions, we have thoroughly revised our manuscript and its final version is enclosed. Point-by-point responses to the comments are listed below.

Reviewer 4:

This review is interesting, but some sections or sentences should be added. Major revision should be made before re-submission. This manuscript would be re-considered only when the authors respond all comments.

1-The mechanism or expression protein in macrophage polarization should be indicated.

Response: According to the reviewer’s comment, the mechanism of TAMs polarization was added in the last paragraph of section 2 (2-Diversity, polarization, and function of macrophages/TAMs) and highlighted.

  1. Macrophage cell therapy

Recently macrophage medicine utilizing polarization for cancer therapy (https://doi.org/10.1016/j.jconrel.2023.04.010) or diagnosis (Nature Biotechnology volume 37, pages531–539 (2019)), respectively named as MacTrigger and Macrophage sensor. Utilizing macrophage polarization is one of the most important factors in using macrophage. These methods are capable of effect switching under TME. The authors should describe these papers.

Response: According to the reviewer’s comment, the mentioned papers were described in the last paragraph of section “5-4 Macrophage cell therapy” and highlighted.

  1. The authors should mention the nanoparticles for TAM target.

Response:  According to the reviewer’s comment, the section “5-6 Nanoparticles Targeting Macrophages” was added to the manuscript and highlighted.

Reviewer 5 Report

Please see general and specific comments included in word file.

Please see general and specific comments included in word file.

Author Response

Dear Reviewer,

Thank you very much for your prompt attention to our manuscript. We are very pleased to receive your positive message and we appreciate the opportunity to be able to respond to your comments.

We hope that you will be satisfied by our responses and that you will now be able to accept the paper for publication.

In this review entitled “Macrophages -based therapeutic strategies in Hematologic Malignancies”

authored by Khalili et al., authors aim to provide a comprehensive overview on diverse activities of macrophages/TAMs at different stages of tumor development and discuss therapeutic strategies targeting TAM and TAM-linked processes.

The review article was divided into 8 sections and some subsections. The main sections were listed here:

  • Introduction
  • Diversity, polarization, and function of macrophages/TAMs
  • Pleitropic activities of TAMs
  • Macrophages and Cancer development
  • TAMs as a therapeutic target
  • Macrophage and hematologic malignancies:
  • Challenges in TAMs-based therapeutics (in cancers or hematologic cancers)
  • Future remarks

This article division reflects very nicely diverse aspects of macrophage biology, which is very much appreciated. This review is also supported by 5 figures and 2 tables, and 317 references.

The reference selection is very thoughtful, and precisely reflect the last development in this area of research. Besides of including peer-review articles, authors also included some conference reports, which did add an additional value to this review article.

This is a very complex and comprehensive review, on which I can congratulate authors for their very hard work.

However, to improve shape of the draft several changes must be made. I sincere recommend sending this draft to scientific English editor for improvement of style and language. Also, further improvement of conciseness would be needed, as specific sections are written in completely different style.

Authors also should check all abbreviations and expand them as recommended.

Response: Thank you for your consideration. We checked all abbreviations and expanded them.

Please ask also scientific English editor to check the manuscript for punctuation, grammar, typos. In several places words are not separated.

The manuscript was re-checked for punctuation, grammar, typos.

In several places also phrases are not starting with capital letter, or in few places two commas or dots are left.

The manuscript was re-checked and all typing mistakes were corrected.

Please also make sure that the entire draft has the same style, and same size, specific section titles are inconsistently bolded.

The style of draft was checked

Authors should also include phrases to bridge specific paragraphs together.

Thank you for your consideration; it was corrected.

Also please make sure that references have the same style in bibliography.

Thank you for your consideration; it was corrected.

Conference reports should be annotated with conference name, year, conference book, journal -please review and correct accordingly.

While there is a decent amount of work to do, I am confident that this will help to improve shape of the draft and would be endorsed upon addressing these comments for publication.

Besides of general comments please find some specific comments below:

Line 35 in abstract-perhaps environment would be better than context

Thank you for your consideration; it was corrected.

Line 38- subpar sound little bit too mild- perhaps “limited” or “ reduced” would reflect the events better?

Thank you for your consideration; we corrected all of them and highlighted them with blue color.

Line 41- focused seems to be more adequate

Line 44-45- please link elegantly these two sentences to make better flow in abstract

Authors might also need to rewrite sentence in line 45 to get better flow with the following sentence.

Line 45- would rather write myeloid cells than myeloid. Perhaps even better to write just TAMs as the review is about TAMs , otherwise it sounds again too general.

Line 47-please use better “ aims” than “ tries” it sounds more confident and stronger

Line 49- please combine this sentence with previous one and try to finish the abstract stronger.

Graphical abstract:

Please explain why do you mention only process of metastasis, while in abstract you want to address mainly hematologic malignancies? Perhaps tumorigenesis or tumor development would be more universal?

Line 66- change cancers to tumors-will be more general.

Thank you for your consideration; we corrected all of them and highlighted them with blue color.

Line 78- please use aim instead of “try”, and cancers to tumors or malignancies

Line 85-please indicate the reason of introducing LAMs at this stage of review.

Line 88-please indicate when macrophages are known or described as TAMs.

Thank you for your consideration; we corrected all of them and highlighted them with blue color.

On Page 4 to page 6 the numbering of lines is missing….the draft is written in different fonts and sizes.

The font and size of all manuscript were checked (Times new roman 12 for text and Times new roman 10 for tables and reference list)

Page 4 last sentence- please revise as the sentence

’On the other hand, M2-like TAMs, which are alternatively activated and induced by Th2 cytokines (such as interleukin-4, interleukin-10, interleukin-13, and/or M-CSF), can promote tumor growth and TME remodeling through production of immunosuppressive and growth factors, proteases, and pro-angiogenic molecules (16).´please clarify if M2 macrophages are only activated by Th2 cytokines? This info seems to be not complete.

Response: More recent research has revealed that macrophage polarization is a much more complex and dynamic process, influenced by various factors within the local microenvironment. It is now recognized that M2 macrophages can be activated by a broader range of stimuli beyond Th2 cytokines. The activation of M2 macrophages can be triggered by IL-10, IL-33, transforming growth factor-beta (TGF-β), immune complexes, glucocorticoids, and other molecules.

Line 92- which TAMs authors meant?

Overall, phenotypic switches in TAMs depends on the combinational microenvironmental factors mainly the action of hypoxia and cytokines availability. In addition, the lack of an approved strategy in measuring intratumoral hypoxia still represents main obstacle for better characterization of hypoxic TAMs. For example, M1 phenotype is generally generated in LPS mediated hypoxic responses while hypoxic TAMs are more strongly associated with M2-like response.

Is hypoxia only contributing to TAM repolarization?

Response: we added other factors

This section lacks a short summary and lacks link to next section.

In this section, the general diversity, intrinsic regulation of macrophage polarization and their functions were introduced. At the end of this section, a concluding paragraph is added and in the next section, the pleiotropic activities of TAMs in tumors and their probable mechanisms in cancer development are discussed.

Line 97 section 3-please correct typo in title: Pleitropicà Pleiotropic

Authors wrote next: “The center, front (the boundary between the tumor cells and the stroma), and the stroma of tumors are the three main locations where macrophages can be found (20). “ please define what is center, what is front etc.

The paragraph was edited for better understanding. “Macrophages are specialized to function in specific microenvironments, which contributes to their location in tumor tissues  [19]. The tumor stroma, the tumor center, and the boundary between the tumor cells and the stroma named invasive front are the three main locations where macrophages can be found [20]. It has been indicated that the distribution pattern and distinct functions of macrophages may be related with different cancer progression mechanisms and the location-related signals they receive“

Further after sentence accompanied with reference 21, authors wrote : “M2 phenotype macrophages may be preferentially involved in promoting the movement of cancerous cells in the invasion zone, facilitating metastasis in stromal and perivascular areas, and stimulating angiogenesis in avascular and peri-necrotic hypoxic areas” but they don’t mention upon what signals mentioned in previous sentence these changes occur. Please review this and change accordingly.

This part was reviewed and corrected as bellow “for example, in colorectal cancers, M2 phenotype macrophages may be preferentially involved in promoting the movement of cancerous cells in the invasion zone, facilitating metastasis in stromal and perivascular areas, and stimulating angiogenesis in avascular and peri-necrotic hypoxic areas via induction of S100A8 and S100A9 (calcium-binding proteins correlated with differentiation and metastasis).“

Thank you for your consideration; we corrected all of them and highlighted them with blue color.

Line 99-100-please revise- this kind of statement is repeating few times throughout the entire draft. Repetitions are redundant , please revise and adjust the text.

The text was adjusted

Line 100-101 this statement was also already previously mentioned.

The text was adjusted

Line 101-102- what do author mean by “trivial outcome” please explain.

Line 102-103- is repetition.

Line 104-105- is repetition of the section above.

The text was adjusted

Line 108- please advise the meaning of attending of M2 macrophages/revise.

Line 99-112- please improve conciseness

Line 115-117 please make the font consistent

Fig. 1 please expand the legend as some terms and abbreviations included in Fig 1 are not explained in section about inflammation.

Line 162- please explain what is HIF1

Thank you for your consideration; we corrected all of them and highlighted them with blue color.

 “The transcription factor HIF1α (hypoxia-inducible factor α) is an oxygen-dependent transcriptional activator consistently observed in macrophages. This factor plays critical roles in the angiogenesis of tumors, regulates the response to hypoxic stress (by switching from aerobic to anaerobic metabolism) … .”

Line 163-please explain what is Ccl2, 9 164 Cxcr4, please make sure that names of cytokines, receptors and genes are in line with guidelines for receptor, genes, proteins nomenclature.

Line 171- what cells are mentioned as these? Macrophages or pericytes? Its not clear from context

Line 173- please explain what is Tie-2

Thank you for your consideration; we corrected all of them and highlighted them with blue color.

 “Tie 2 is a tyrosine kinase receptor with critical function in vascular stability.”

Line 181- please indicate if activated classically or alternatively

Line 188 – there is discontinuation of lines numbering.

Thank you for your consideration; we corrected all of them and highlighted them with blue color.

Line 189- perhaps explanation what is the difference between tumor niche and pre-metastatic niche would be of importance.

According to the reviewer’s suggestion, the mentioned difference was added and text was edited as below “The tumor niche is as a specialized environment including infiltrating immune cells, tumor cells, activated stromal cells, and extracellular matrix regulating tumor progression while the pre-metastatic niche is an essential requirement for the colonization of circulating pro-tumor cells in particular organ and metastasis of primary tumors. Macrophages are one of the most remarkable immune cells within pre-metastatic niches.“

Thank you for your consideration; we corrected all of them and highlighted them with blue color.

Line 193/194- please explain why CAR-M is discussed in the middle of section related to tumor metastases? This section would rather fit into therapeutic approaches.

Line 205- again please explain what is MAM.

Line 222- AAM endotypes -please expand and explain

Thank you for your consideration; we corrected all of them and highlighted them with blue color.

Alternatively activated (M2) macrophages

Line 255-256- “The gut microbiota can stimulate the production of ROS during intratumoral phagocytes, which enhances the effectiveness of these compounds.”  Please revise these two last phrases, its hard to understand the flow of thoughts.

Thank you for your consideration; we edited it.

Line 260-molecule is not antibody

Line 260-261-please be specific, what cells are recruited etc-its little bit too general.

Fig. 3 please provide legend to this figure, again the schematic is not self-explaining. What should be the link between M2/M1 macrophages and immunotherapy progression?

Line 301-302-please explain what association was described.

Line 303-376-authors provide lots of different strategies, however this require an additional effort to link these strategies into one paragraph of logic and smooth transition from one strategy to another one. This has to be revised.

Line 377-398-the section addressing the role of peptides is very hard to understand. Suggest to restructure.

Table 1- based on content the table is not reflecting the role of macrophages in treatment, please change title or adjust content to title.

Alternatively given the fact that the content addresses only research in solid tumors, perhaps authors could modify the Table 2 and in similar way show changes in hem malignancies.

Line 406-424-given the fact that authors don’t discuss solid tumors, it would be rather appreciated to merge these lines together. Please also shape this intro from general to specific information.

Line 423- seem to be redundant.

Line 426- please indicate if authors did mean activation or inhibition of the CXCR4/CXCL12 axis.

Line 434-does control group supports M2 polarization? What was control group? This is not clear from the context, please revise.

The ALL section is style wise different than AML section, please unify style.

Thank you for your consideration; we corrected all of them and highlighted them with blue color.

Please indicate the link between sentence starting in Line 468 and later in Line 470

Line 475 and further- please clarify what kind of killing authors did authors mean?

Line 478-please explain what the experimental group was.

Line 485- what cells authors meant? And what synergistic effect authors meant? Please clarify.

Please indicate if there is any evidence of CLEC12A on macrophages? Is the expression level meant on macrophages or AML cells? If on macrophages , than on which type of macrophages?

Line 492- please explain what exactly type of macrophages is it?

Authors use TAM, M2, Mφs and LAMs without explaining the crucial differences between these two last groups, please correct and apply changes.

Please unify the terminology or explain upfront , this would help readers from non-hematological field easier comprehend such complex topic.

Line 511- authors use names of compounds without explaining what these targets, please correct it.

Thank you for your consideration; we corrected all of them and highlighted them with blue color.

Line 522- NLCs-please explain/expand

It was explained before (section 6.1. “Nurse-like cells (NLCs), also known as CLL-specific TAMs expressing CD68 and CD163[264]”

Line 530- expand what is TNFR

Thank you for your consideration; we corrected all of them and highlighted them with blue color.

.”Tumor necrosis factor receptors (TNFRs) are membrane proteins that activate cell death”

Line 538- NLS should be explained above.

Please revise and combine info about NLC into one concise section.

Also, short summary for either each type of malignancy or together for all hem malignancies would be appreciated.

Line 552- induced or supported?

Line 556- poorer than what?

Line 556 /557-please explain as these two following sentences are contradictory

Line 559- please explain in which way are these markers biomarkers of response.

Line 560-please be consistent…LAMs are here not explained….plus again the one sentence is contradictory to previous one- please revise this section.

Line 565- in T-cell non-Hodgkin lymphoma- seems like “of” is missing

Line 572/573- please correct this sentence “Enhanced M2 macrophage and activation lipid metabolism has been observed in the immunosuppressive tumor microenvironment of non-MYC/BCL2 double express or DLBCL (269)”

Thank you for your consideration; we corrected all of them and highlighted them with blue color.

Line 581- This process increases the production of CXCL13 and activates 581 osteoclasts, which enhances their ability to resorb bone and promotes MM progression.- could authors please correct this sentence?

I guess CXCL13 but not osteoclasts enhance their ability to resorb bone etc….

Line 583 – this sentence is repetition of line 576 where authors wrote that TAMs are prevalent…..

Line 586- please expand what is PFKFB3-JAK1 signaling axis

Line 590- please explain what is RGS12.

Line 592-please expand what is BMME. The full name was mentioned before in page 26 line 511

Line 591/592-Authors refer to Fig 4 but there is no explicit indication that the schematic Figure can be applied also to MM, please correct both text and Fig 4 in such case.

Thank you for your consideration; we corrected all of them and highlighted them with blue color.

Fig 4

Needs explanation of used abbreviations for cytokines.

The indicated functions of M1 and M2 macrophages are not exactly clear.

From the figure for instance readers would assume that in leukemia macrophages polarize to M1 that secrete lots of cytokines associated with tissue damage or microbial or tumor activity? Please add legend and please clarify descriptions as it transfers wrong information.

Thank you for your consideration; we corrected all of them and highlighted them with blue color.

7- Challenges in TAMs-based therapeutics (in cancers or hematologic cancers)

Please change the title- to (in solid or hematologic tumors)-cancers only originate from epithelial cells.

Line 602- perhaps instead increase infections -authors could write : increase risk of infections would be more accurate.

disorganize tissue-resident cells to achieve their usual functions-please explain what is meant by “ disorganize”.

Line 603- “Therefore, discovering TAM-specific molecules or markers that are mainly created through metastasis associated macrophages (MAMs) and/or activated M2 (AAMs) will enable enlightened  therapeutic approaches that could specifically target tumor cells without affecting the normal function of other immune cells which are tissue-resident (277).”-could authors explain what is meant by enlightened?

Thank you for your consideration; we corrected all of them and highlighted them with blue color.

If MAMs or AAMs are not mentioned before there is no need to add abbreviations.

The aspects mentioned in these sections are of importance but need to be little bit better structured.

Perhaps starting with targetingà than preserving the physiological functionà and blocking TAM/Cancer cell interaction would help. Please apply these changes for better flow of reading.

Thank you for your consideration; we corrected all of them and highlighted them with blue color.

8-future remarks-

Please add 1-2 sentences addressing what strategy in opinion of authors has the strongest potential to boost the efficacy of anti-TAM therapy in future and what directions ate still underdeveloped and should be more investigated in the future.

Table 2:

The second row of the table- ( please include the name of the inhibitor, if available) is the same inhibitor as mentioned din row 3? If yes please correct accordingly.

The next row with Lenalidomide- please complete description in column mechanism observation…as it seems to be cut.

In  row ALL Anti-CD47 mAb ( please provide the name of Ab if available)

Same in row below CD204-positive TAM-please provide the name of Ab If available

BMP4-please verify if expected mechanism of this compound/agent should indeed work through Inducing immunosuppressive dendritic cells and favoring the generation of M2- like macrophages with pro-tumoral features????? Intuitively it should be opposite?

Exposure to myeloid differentiation promoting cytokines-please indicate what cytokines/ single cytokines? Or cocktails of cytokines?

In clinically tested compounds/Ab please provide NCT number.

Please verify if the anti-CD47 compound  works only through requirements- and how this corresponds to key mechanism on macrophages? As this is not mentioned in table.

Here perhaps as an example of side effects/adverse effects would be important to include separate column? Or in same column specify mechanism and eventually adverse effects.

Please complete with what drug is Brentuximab conjugated for completeness.

Some of agents are repeated for instance Anti-CD47 mAb Hu5F9-G4 Clinical / Phase II Enabled phagocytosis of tumor cells by TAM-perhaps this can be combined with first time mentioned anti-CD47 mAb?

Thank you for your consideration; we corrected all of them and highlighted them with blue color.

Please keep description of drug actions/mechanism consistent. For instance, once authors describe-inhibition, another time inhibitor etc.

In section regarding myeloma- please correct typo: Atiangiogenic and macrophage killing due to CCL2-CCR2 signaling axis

Perhaps division of table into:

Colum with drug name/target/mechanism of action/adverse effects would help to clarify some inconsistencies.

Also, separate column indicating if the drug/ab is preclinical or clinical would help.

Thank you for your consideration; we corrected all of them and highlighted them with blue color.

Round 2

Reviewer 4 Report

OK.

good